# Effects of Defective Unloading and Recycling of PCNA Revealed by the Analysis of *ELG1* Mutants

**DOI:** 10.3390/ijms24021568

**Published:** 2023-01-13

**Authors:** Ziv Itzkovich, Karan Choudhary, Matan Arbel, Martin Kupiec

**Affiliations:** The Shmunis School of Biomedicine and Cancer Research, The George S. Wise Faculty of Life Sciences, Tel Aviv University, Tel Aviv 69978, Israel

**Keywords:** genome stability, *Saccharomyces cerevisiae*, DNA replication, DNA repair, mutagenesis, *SRS2*, gene silencing

## Abstract

Timely and complete replication of the genome is essential for life. The PCNA ring plays an essential role in DNA replication and repair by contributing to the processivity of DNA polymerases and by recruiting proteins that act in DNA replication-associated processes. The *ELG1* gene encodes a protein that works, together with the Rfc2-5 subunits (shared by the replication factor C complex), to unload PCNA from chromatin. While *ELG1* is not essential for life, deletion of the gene has strong consequences for the stability of the genome, and *elg1* mutants exhibit sensitivity to DNA damaging agents, defects in genomic silencing, high mutation rates, and other striking phenotypes. Here, we sought to understand whether all the roles attributed to Elg1 in genome stability maintenance are due to its effects on PCNA unloading, or whether they are due to additional functions of the protein. By using a battery of mutants that affect PCNA accumulation at various degrees, we show that all the phenotypes measured correlate with the amount of PCNA left at the chromatin. Our results thus demonstrate the importance of Elg1 and of PCNA unloading in promoting proper chromatin structure and in maintaining a stable genome.

## 1. Introduction

DNA replication is one of the central processes of life. The robust and fast replication of the entirety of the genome is essential for life and must be well regulated. If the replication takes too long, the cell cycle may become disrupted; a fast but inaccurate DNA replication, on the other hand, would lead to accumulation of mutations, most of which will be deleterious to the next generation [1,2]. Thus, a well-kept balance exists between the accuracy and the speed of replication in all organisms investigated. This balance is maintained by a group of proteins involved in DNA replication, DNA damage avoidance, and DNA damage repair [3,4,5]. The full and accurate replication of the genome poses another challenge, which is the unpacking and repacking of DNA during replication [6]. The genomic DNA is neatly organized within the nucleus [7,8]. DNA is wrapped around nucleosomes, and post-translational modifications of the histones that compose these basic structures determines whether chromatin is in a (mostly) active (euchromatin) or inactive (heterochromatin) configuration [9,10]. Since DNA replication requires the opening of double-stranded DNA, nucleosomes must be evicted from DNA. In order to maintain the same epigenetic state, the two newly replicated DNA molecules have to be repacked around nucleosomes containing the same histone modifications present before DNA replication. This is achieved by the activity of a large number of proteins involved in the transfer of modified histones to both leading and lagging strands on the wake of the moving fork [11]. The activity of all these important factors is coordinated and regulated by the DNA replication clamp, PCNA [12,13,14].

PCNA is a homotrimeric complex that acts as a sliding clamp, enclosing DNA [15]; it is a master regulator for different DNA repair and chromatin assembly processes [16,17]. PCNA coordinates the work of many proteins involved in many cellular processes. PCNA functions as a processivity factor for DNA polymerases [18]; it helps recruiting Okazaki fragment maturation factors such as Rad27 and Cdc9 (FEN1 and LIG1 in humans) [19,20]; it is involved in the recruitment of DNA damage tolerance proteins such as Rev3 and Rad5 [21,22]. In addition, PCNA has physical interactions with proteins involved in histone deposition and chromatin assembly at the fork, such as the CAF-1 subunit Cac2 [17]. Thus, understanding the regulation of PCNA and the dynamics of its loading and unloading from chromatin is of utmost importance.

PCNA is recruited and loaded onto chromatin by the RFC complex [23], composed of a large subunit (Rfc1) and four small subunits (Rfc2-5). Three additional RFC-like complexes (RLCs) share the small subunits, but each one carries a different, unique protein instead of Rfc1. Whereas one of them (carrying the Rad24 protein in *S. cerevisiae* and RAD17 in humans) oversees the loading of the 9–1–1 checkpoint complex, the two other RLCs, carrying Elg1/ATAD5 or Ctf18/CHTF18, interact with PCNA. The focus of the current study is the Elg1 RLC, which has been shown to work as the main unloader of PCNA (reviewed in [13]). Elg1 is a nonessential protein, and its deletion causes a wide range of phenotypes, including increased chromosome loss and homologous recombination, [24], increased sensitivity to DNA damage [25], and defects in chromatin maintenance [26]. In addition to this, *ELG1* also genetically interacts with the genes encoding the sister chromatid cohesion machinery [27,28]. Over the years, there has been much debate on the possible roles of Elg1, and whether it carries any role outside of PCNA unloading, as hinted by its importance in many cellular processes. Here, we take advantage of the availability of a collection of *elg1* mutants to probe their effect on various phenotypes. Our results suggest that all the phenotypes of *elg1* mutants are due to the accumulation of PCNA on the chromatin, suggesting that Elg1’s sole role is the timely unloading and recycling of PCNA.

## 2. Results

### 2.1. A Collection of elg1 Mutants

Deletion of the *ELG1* gene is not lethal, but results in a vast array of phenotypes, including sensitivity to DNA damaging agents, hyper-recombination, chromosome loss, and decreased chromatin silencing (reviewed in [13]). However, most experiments were carried out with strains bearing a total deletion of the *ELG1* gene. To investigate whether all the phenotypes of *elg1* mutants were due to a defect in PCNA unloading activity, we created a collection of point mutations in the *ELG1* gene, which affect Elg1’s unloading activity to different degrees, and studied their effect on a number of phenotypes. We created mutations at different motifs in the protein, as follows (Figure 1):(1)A sequence at the N-terminus of Elg1 bears resemblance to the PCNA-interacting peptide (PIP) of Rfc1 [29]. We mutated amino acids N55 and S57 to alanine (*elg1-NS55,57AA*; this allele is hereafter referred to as *elg1-PIP*).(2)Three SUMO-interacting motifs were found in the same region, and point mutations in them (I28A I93K II121/2AA) reduced the interaction between Elg1 and SUMO [29]. We refer to this allele as *elg1-SIM.*(3)The third region mutated in the N terminus was an unstructured loop region, spanning from aa 290 to aa 319 and containing a hydrophobic patch. This site is a possible protein–protein interaction site and is conserved throughout all Elg1 orthologs but is absent in other components of the RFC complex and in the other Rfc1-like proteins [30]. We deleted this loop and replaced it by a short peptide. This allele is referred to as *elg1-loop*.(4)By homology modeling of the Elg1 protein on the solved crystal structure of the RFC complex together with PCNA, it was determined that threonines 386/7 in Elg1 correspond to asparagines 694 and 695 of the human Rfc1, which were shown to be at the interface of the two proteins [30]. These residues were mutated to either aspartic acid (hereafter referred to as *elg1-TT386/7DD*) or to alanine (*elg1-TT386/7AA*).(5)These mutants were also combined with the SIM mutation, creating *SIM+AA* and *SIM+DD* alleles.(6)Elg1 and all members of the AAA+ family contain Walker A and Walker B motifs [31]. The current models suggest that ATP binding is crucial for the association of Elg1-RLC with PCNA for its unloading, and that the hydrolysis of ATP is important for the subsequent detachment of Elg1-RFC from PCNA, allowing its recycling [23,32]. We mutated the two lysines (at positions 343, 344) either to aspartic acid (*elg1-KK343/4DD*) or to alanine (*elg1-KK343/4AA*). These are referred to hereafter as *WalkerA-DD* (*WA-DD*) and *Walker A-AA* (*WA-AA*).(7)We also mutated two aspartic acids at positions 407,409 that lie within the Walker B motif, which are conserved between Elg1 and its human ortholog *ATAD5* [23]. These residues are thought to be crucial for ATP binding and/or hydrolysis. They were mutated either to lysine (*elg1-DD407/409KK*) or to alanine (*elg1-DD407/9AA*), hereafter referred to as *Walker B-KK* (*WB-KK)* or *Walker B-AA (WB-AA*).(8)We also created a double Walker A + Walker B mutant: *elg1-KK343/4DD+DD407/409AA*, hereafter referred to as *elg1-WAB.*(9)Finally, we mutated position 390 from valine to aspartic acid (*elg1-V390D*) and to alanine (*elg1-V390A*). These mutations disrupt the similarity in Elg1 at positions 381-390 (LLDFTTTHYV) to a small patch of the DNA repair and telomeric protein Yku80 (aa 450-456 LLDrTTTsgV). Moreover, the valine is conserved throughout Elg1 orthologs [23].

All the strains carrying the *ELG1* alleles on centromeric plasmids, or replacing the *ELG1* gene in the genome, showed the same cell cycle distribution (Appendix A). We tested our battery of mutants for a number of phenotypes that characterize the full *ELG1* deletion.

### 2.2. Sensitivity to DNA Damage

Methyl methanesulfonate (MMS) is a DNA alkylating agent that impairs DNA replication (as alkylated DNA is poorly replicated by DNA polymerases and must be efficiently repaired) [33]. All *elg1* mutants, as well as the wild-type control, were cloned into a centromeric plasmid (pRS415) and introduced into a *elg1Δ* yeast strain. An empty vector served as an additional control. Serial dilutions of the different strains were spotted on SD-complete plates or on similar plates with varying amounts of MMS (Figure 2). Wild-type cells were able to grow at all MMS concentrations used, while *elg1Δ* cells exhibited increased sensitivity to the DNA damaging agent.

In accordance with previous results [34], no sensitivity to MMS was observed for the *PIP* or *SIM* mutants. Among the different Elg1 mutants, MMS sensitivity was seen for TT386/7 when mutated, especially to aspartic acid, but they also exhibited a mild sensitivity in the highest of MMS doses tested when changed to alanine. This sensitivity was increased when combined with the *SIM* mutations. Mutation at V390 also seems to confer increased sensitivity when mutated to aspartic acid but not to alanine. Growth sensitivity was also seen for both Walker mutations, with the lysines at 343/4 (Walker A region) showing sensitivity when mutated to aspartic acid but not to alanine, and the aspartic acids at positions 407/9 (Walker B region) showed increased sensitivity when mutated to both lysine and alanine.

Apart from the mutations mentioned, no other *elg1* mutation within the group increased MMS sensitivity.

### 2.3. Mutation Rate

The rate of genomic mutation is usually assessed by either measuring the rate of inactivation of a gene that confers sensitivity to a drug (e.g., resistance to canavanine, acquired by inactivating the arginine pump encoded by the *CAN1* gene), or by reversion of a particular auxotrophic mutation. In the absence of the *ELG1* gene, the rate of mutation is not greatly altered at the *CAN1* locus, but slippage mutations are strikingly elevated [35]. These can be easily scored using strains in which a row of 14 adenine residues is inserted into the *LYS2* gene [36]. Addition of a single residue or deletion of two nucleotides restores the reading frame and confers a Lys+ phenotype. *elg1Δ* strains exhibit roughly a ~5 fold increase in the slippage rate using this assay. A possible explanation is that over-retention of PCNA on the chromatin in the deletion mutant results in untimely or reduced activity of the mismatch repair complex [37].

We introduced all the *elg1* mutations into a strain carrying the *lys2-14A* allele and measured, by fluctuation rate, their rate of mutagenesis (Figure 3). The *elg1* mutant panel could be divided roughly in two groups, one that exhibited a wildtype level of mutagenesis and another that had levels similar to the *elg1Δ* allele. The first group includes the *PIP*, *SIM*, *TT386/7AA*, *loop*, *V390A*, and *WA-AA* alleles. Members of the second group are the *TT386/7DD*, *SIM+AA*, *SIM+DD*, *V390D*, *WA-DD*, *WAB*, *WB-KK*, and *WB-AA*.

### 2.4. Rescue of the Synthetic Lethality of Elg1Δ Srs2Δ Double Mutants

The Srs2 helicase plays a central role during the repair of double-stranded breaks by homologous recombination [38] and has been shown to prevent the bypass of genomic lesions using this mechanism [39]. Double mutants *elg1Δ srs2Δ* grow very slowly as haploids and are essentially dead as diploids [40]. The reason for this lethality is still unknown. We used our battery of mutants to ask which of them was able to suppress the lethality. We created a diploid stain carrying genomic deletions of the *ELG1* and *SRS2* genes, which is kept alive by the presence of a *URA3*-marked plasmid bearing the *ELG1* gene. These cells cannot lose the plasmid, as the diploid double mutant without it is inviable. Selection against the *URA3* marker carried by the covering plasmid (on 5-FOA-containing medium) allows us to test whether the *elg1* allele introduced can complement the activity required for survival of the strain. Figure 4 shows that a control plasmid carrying the wildtype *ELG1* gene complements, allowing growth on 5FOA, whereas an empty vector is unable to allow plasmid loss. Again, the *elg1* mutants can be divided in two categories, which coincide with the results observed in the slippage assay above.

### 2.5. Gene Silencing in elg1 Mutants

Regions of the genome that are silenced, such as the mating type cassettes *HML* and *HMR* in yeast, are not expressed, and are maintained in a heterochromatin-like state despite the fact that cells proliferate. Thus, the cells are able to duplicate the genome and to maintain an epigenetic memory of the local chromatin state. We have shown in the past that *elg1Δ* cells are defective in this process, using the CRASH (Cre-reported altered states of heterochromatin) assay [17]. This system, which monitors silencing at *HML*, has two genomic components: the first is a Cre recombinase gene inserted at *HML*. The second component is an RFP gene, placed between two LoxP sites, downstream to a strong constitutive GPD promoter and upstream to a promoterless GFP gene. If silencing at *HML* is lost, even transiently, and the Cre gene is expressed, leading to site-specific recombination between the LoxP sites, it results in a permanent switch in expression from RFP to GFP (Figure 5A). Such silencing-loss events are manifested as green sectors in colonies and green cells in flow cytometry and appear at low levels in wildtype cells (Figure 5B).

Flow cytometry analysis was used to quantify the rate of RFP to GFP switching. Cells that have recently switched express both RFP (remaining from the previous generations) and GFP (newly switched). Thus, the number of both RFP- and GFP-positive cells compared to the total number of cells shows the *apparent* rate of silencing loss. We call this rate “apparent” because RFP molecules may be still present in cells for more than one generation after switching from RFP to GFP expression. However, this calculation allows a quantitative comparison between the different mutants. We tested all the mutants in the CRASH assay, and the results of three experiments for each strain are shown in Figure 5C.

The *elg1Δ* mutant exhibits a 12-fold increase in the apparent rate of silencing, in comparison to the wildtype control. The *PIP*, *SIM*, *loop*, *V390A*, and *Walker A-AA* alleles showed wildtype levels of silencing. The *elg1-TT386/7AA* allele showed a slight but persistent increase that was statistically significant. Addition of the SIM mutations further increased the silencing twofold. Similarly, the *elg1-V390D* allele showed an intermediate level of silencing, whereas the *elg1-TT386/7DD, sim+DD, WA-DD, WB-KK, WB-AA* and *WAB* showed an extent of silencing loss comparable to that of the *elg1Δ* strain.

### 2.6. PCNA Accumulation Levels in the elg1 Mutants

We then proceeded to measure the level of chromatin-bound PCNA levels in all our different mutants. (Figure 6). A wildtype strain and a full *ELG1* deletion served as appropriate controls. All cell cultures were grown to mid-log in rich medium (YPD), and proteins were extracted and subjected to fractionation into chromatin and cytoplasmic fractions (see Materials and Methods). This process was repeated independently three times and the results were quantified after normalizing to the protein levels of the chromatin-bound protein H3 (Figure 6). As the *elg1-sim* mutants had slightly lower expression levels (Appendix A), we overexpressed their protein from high copy number plasmids (Appendix A). However, Appendix A shows that PCNA accumulation was identical irrespective of protein levels, indicating that the phenotypes observed were not affected by expression levels. *elg1Δ* cells exhibited a sevenfold increase in the level of chromatin-bound PCNA in comparison to the wildtype control. The *PIP* and *WalkerA-AA* mutations showed wildtype levels of PCNA. Interestingly, the *Loop* and *V390A* mutants showed lower than wildtype levels (0.85 and 0.88, respectively). The SIM mutant exhibited a 1.7-fold increase, the *TT386/7AA* mutant a 1.9-fold elevation, which was further increased by addition of the SIM mutations to 3-fold. The *TT386/7DD* mutant also showed a 3-fold increase in chromatin-bound PCNA, and this value was further elevated to 4.63-fold by the addition of the SIM mutations. The *V390D* showed a 3.5-fold increase, the *Walker A-DD* a 6-fold elevation, and the *Walker B* and the combined *Walker AB* mutant a 4–5-fold increase in chromatin-bound PCNA.

Thus, all point mutants examined retained *some* level of PCNA unloading, and none reached the accumulation seen in the *elg1Δ* allele. Our mutants showed a gradient of effects on PCNA accumulation, and in general, it is clear that all the phenotypes analyzed correlate well with the amount of PCNA accumulated, although differences could be observed depending on the phenotype scored (see Discussion). The correlation is most clearly seen for the results of the CRASH assay (Figure 5 and Figure 6C), in which subtle phenotypes are observed for mutants with weak effects, but it can also be seen for DNA damage sensitivity (Figure 2), Lys+ mutation rate (Figure 3), and the suppression of synthetic lethality in the *srs2Δ elg1Δ* strain (Figure 4). Our results thus further solidify the notion that Elg1’s PCNA unloading role is the main reason for all the genomic instability effects seen in *elg1* mutant cells.

## 3. Discussion

PCNA is a central player during DNA replication and repair. It acts as a landing platform that directs proteins to the moving replication fork, according to need. The proteins recruited may act on DNA (e.g., DNA polymerases, Okazaki fragment maturation enzymes) or may affect chromatin remodeling (e.g., histone chaperones). The combined work of loading PCNA by RFC and its unloading by the Elg1-RLC provides the cell with the ability to recycle PCNA quickly and accurately to and from needed areas during replication.

Whereas no subunit of the RFC complex can be deleted without losing viability, cells carrying a full deletion of the *ELG1* gene are alive, but present striking phenotypes related to genome stability: hyper-recombination, gross chromosomal rearrangements, chromosome loss, elongated telomeres, increased loss of silencing at heterochromatic regions, increased mutagenesis, and sensitivity to DNA damaging agents (reviewed in [13]). Previous work has demonstrated that the phenotypes of *elg1Δ* related to sensitivity to DNA damage [34,41], hyper-recombination [24], and telomere length ([42], our unpublished results) were due to the accumulation of PCNA, and in particular its SUMOylated version, on chromatin. However, other processes affected by Elg1 activity, such as the maintenance of gene silencing, the synthetic lethality in the absence of Srs2, or the rate of replication slippage, were not analyzed.

Here, we have presented a panel of *elg1* mutants that display different amounts of PCNA on the chromatin. We have systematically analyzed their performance in several assays: sensitivity to MMS, slippage mutagenesis, synthetic sickness with *srs2Δ*, and loss of gene silencing. From our results, it is clear that all the phenotypes tested are correlative with the amount of PCNA left on the chromatin. Mutants that had no effect on the level of PCNA, such as *elg1-PIP* and *WalkerA-AA*, showed wildtype phenotypes in all assays, whereas mutants that accumulated PCNA to 2/3 the level of the deletion allele (e.g., *elg1-SIM+DD, Walker A-DD, Walker B* alleles, and the double Walker A-Walker B mutant) all showed phenotypes indistinguishable from those of the *elg1Δ* allele. Thus, a ~3–4-fold accumulation of PCNA on the chromatin is sufficient to confer all the phenotypes seen in strains deleted for the *ELG1* gene.

Especially interesting are alleles with intermediate levels of accumulation of PCNA. These resulted in a wildtype or defective phenotype, depending on the assay. For example, the *elg1-TT386/7AA* allele, which shows a 1.87-fold increase in the level of PCNA, has wildtype levels of mutagenesis, shows no synthetic sickness when combined with *srs2Δ*, but shows a very mild sensitivity to MMS, and a slight elevation in the level of silencing loss. When combined with the SIM mutations, PCNA levels rise to ~3-fold of the wildtype cells; this confers only intermediary levels of silencing loss, but is enough to confer mutagenesis, MMS sensitivity, and synthetic lethality to *srs2Δ* to the same extent as the *elg1Δ* allele. Thus, a small difference in the level of PCNA can sometimes make a large difference in the final phenotype.

Why does accumulation of PCNA lead to defective phenotypes? We propose that PCNA that was loaded onto chromatin but was not unloaded by the Elg1 RLC does not get recycled; this in fact results in a shortage of PCNA, required for recruiting other proteins to the moving fork. One such example is the chromatin remodeler CAF-1, which has a known interaction with PCNA [43,44]. We have previously shown that overexpression of CAF-1 can correct the increased silencing loss of *elg1Δ* mutants [17].

We created mutations at the valine residue 390 of Elg1 to test the possibility that a small motif shared with the Yku80 protein may have functional implications. However, the results obtained were very different if the valine was mutated to alanine, or to aspartic acid. In the first case, the strain behaved as a wildtype in all assays, whereas in the second case it was indistinguishable from an *elg1* null. As the protein levels of the two mutants are undistinguishable from those of the wildtype, the different results imply a structural change introduced by the aspartic acid, rather than an inactivation of an important residue in Elg1. Since the valine 390 residue is close to the threonines 386 and 387, which are predicted to interface with PCNA, the addition of the more bulky aspartic acid may make the interaction with PCNA sterically difficult.

A similar result was obtained for our Walker A mutations. When mutating the non-canonical lysine to alanine, there was no detectable phenotype in any of our experiments, but when mutating it to aspartic acid, it mimicked *elg1Δ.* This is in contrast to Walker B mutations, which consistently produced results similar to those of the *elg1Δ* allele. This stands in line with the current model, in which Walker A is important for ATP binding, and the conserved residue does not hold any catalytic activity, while Walker B is important for the hydrolysis of the ATP molecule, and thus does hold a catalytic role and could not be easily mutated without yielding a phenotype [23,31].

To summarize our results, we see a strong correlation between the amount of PCNA left at the chromatin in *elg1* point mutants, and the degree of their phenotype. This supports the notion that all the phenotypes measured are due to improper PCNA unloading and recycling.

## 4. Materials and Methods

Standard yeast media and methods were used. Standard yeast molecular biology techniques were used to create the mutant collection.

### 4.1. Chromatin Fractionation Assay

Cells from 50 mL cultures (OD600 < 1.0) were collected by centrifugation, successively washed with ddH2O, PSB (20 mM Tris–Cl pH 7.4, 2 mM EDTA, 100 mM NaCl, 10 mMb-ME), and SB (1 M Sorbitol, 20 mM Tris–Cl pH 7.4), and transferred to a 2 mL Eppendorf tube. Cells were suspended in 1 mL SB, 30ul Zymolase 20T (20 mg/mL in SB) was added, and samples were incubated at 30 °C with rotation until >85% spheroplasts were observed (60–90 min). Spheroplasts were collected by centrifugation (2 K, 5 min, 4 °C), washed twice with SB, and suspended in 500 mL EBX (20mM Tris–Cl pH 7.4, 100 mM NaCl, 0.25% Triton X-100,15 mM-ME + protease/phosphatase inhibitors). TritonX-100 was added to a 0.5% final concentration to lyse the outer cell membrane, and the samples were kept on ice for 10 min with gentle mixing. The lysate was layered over 1 mL NIB (20 mM Tris–Cl pH 7.4, 100 mM NaCl, 1.2 M sucrose, 15 mM-ME + protease/phosphatase inhibitors) and centrifuged at 12 K RPM for 15 min, at 4 °C. The supernatant (cytoplasm) was discarded. The glassy white nuclear pellet was suspended in 500 uL EBX and Triton X-100 was added to a 1% final concentration to lyse the nuclear membrane. The chromatin and nuclear debris were collected by centrifugation (15 K, 10 min, 4 °C). Chromatin was suspended in 50 uL Tris pH 8.0 for Western blot analysis (Chromatin). To each fraction, an equal volume of 2 × SDS-PAGE loading buffer (60 mM Tris pH 6.8, 2% SDS, 10% glycerol, 0.2%bromophenol blue, 200 mM DTT) was added; samples were incubated at 95 °C for 5 min and were then analyzed by SDS-PAGE and Western blot analyses.

### 4.2. Quantification of Silencing Loss Using Flow Cytometry

For each CRASH strain, 10 single colonies were inoculated separately into 2 mL of SD-Trp-Leu medium in 96-well deep-well plates (VWR) and were grown overnight to saturation at 30 °C on a microplate orbital shaker (VWR). Overnight cultures were diluted in 1 mL of fresh medium at a density of 10^5^ cells/mL in 96-well deep-well plates and were grown at 30 °C on a microplate orbital shaker until mid-log phase. For each culture, a minimum of 50,000 events were collected using an MACSQuant^®^ flow. Gating was used to include only unbudded and budded cells and to measure separately the number of GFP+ cells and the number of RFP+ cells. A Boolean logic gate was used to determine the number of cells that were both GFP- and RFP-fluorescent. The apparent rate of silencing loss is calculated as the number of cells that are both GFP- and RFP-fluorescent divided by the total number of cells.

### 4.3. Staining for Flow Cytometry

Mid-log cells were pelleted and resuspended in 140 µL of ice-cold ethanol 100% and 60 µL of TE, and incubated overnight at 4 °C. The following day, cells were arranged in a 96-well plate, centrifuged at 4000 RPM for one minute, and the supernatant was disposed of. Cells were resuspended in 200 µL of TE with 0.5 mg/mL of RNaseA, and the plate was incubated overnight at 37 °C. Cells were again centrifuged at 4000 RPM for one minute, and the supernatant was disposed of. Cells were resuspended in TE with 0.5 mg/mL of proteinase K solution and incubated for 4 h at 37 °C. Cells were again centrifuged at 4000 RPM for one minute, the supernatant was discarded, and the pellet was resuspended in TE solution with 20 µg/mL of propidium iodide. Cells were then incubated overnight at 4 °C in the dark. The following day, single cells were de-clumped using three rounds of sonication at 20% (for 5 s), then readings were taken using the MACSQuant^®^ flow cytometer machine.

### 4.4. Mutation Measurement

Mutation levels were measured in E134 by fluctuation tests as described [45]. Standard deviations were usually <5%. Each experiment was repeated at least three times.

## Figures and Tables

**Figure 1 ijms-24-01568-f001:**
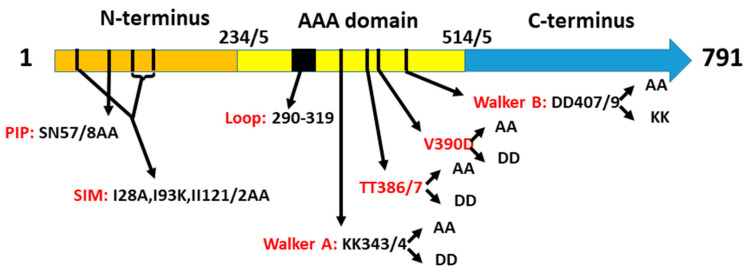
**Schematic representation of the Elg1 protein showing domains and mutation sites.** The *ELG1* gene was divided into three domains. Mutations described in the text affect the N-ter and AAA domains. All proteins are expressed at similar levels.

**Figure 2 ijms-24-01568-f002:**
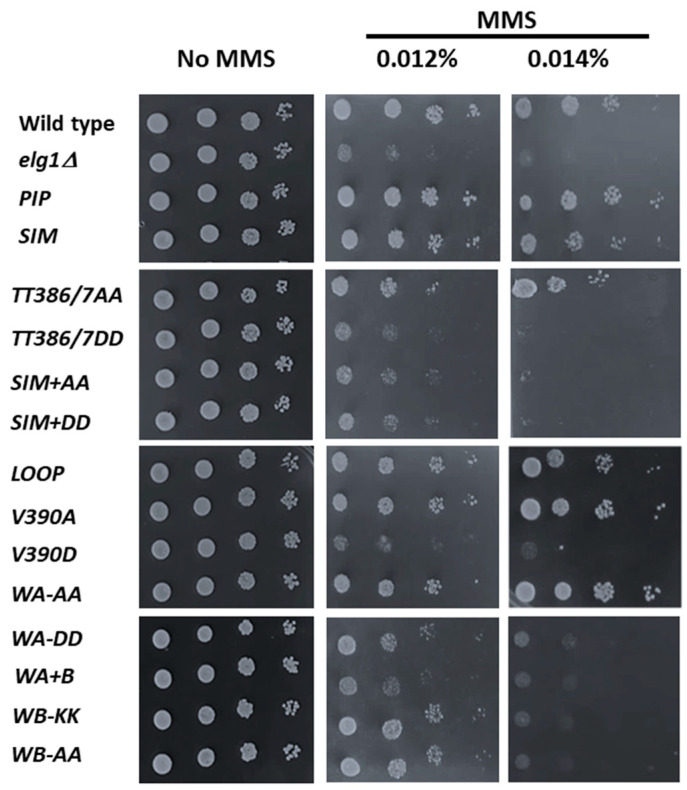
**DNA damage sensitivity assay for different *ELG1* mutations.** Serial fivefold dilutions of yeast cultures on SD-complete without and with methyl methanesulphonate (MMS) at the indicated concentrations show different sensitivities of *elg1* mutants. *elg1Δ* strains were transformed with a centromeric pRS415 plasmid collection containing different *ELG1* mutations. The strain described as “Wild type” carried the *ELG1* gene, and the one described as “*elg1Δ*” carried an empty vector.

**Figure 3 ijms-24-01568-f003:**
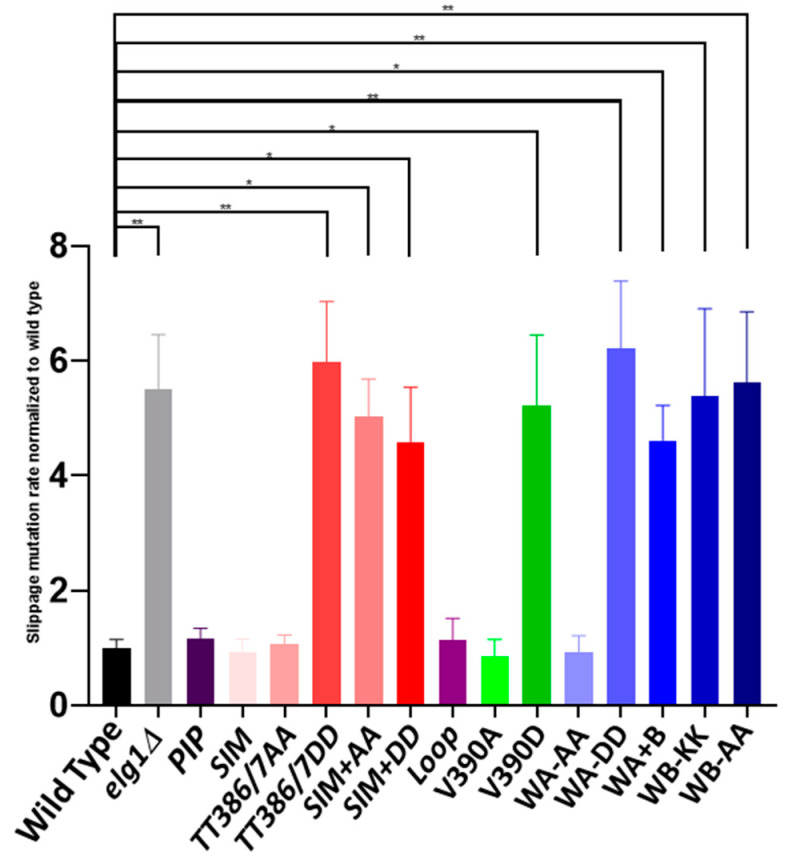
Slippage mutation rates of *elg1* alleles. Reversion of the *lys2-14A* allele was measured by fluctuation test in all isogenic *elg1* mutants. The rates were compared to that of the wildtype, which is considered as 1. At least three experiments were used for quantitation and the error bars represent the standard error of the mean. * 0.05 > *p* > 0.01; ** 0.008 > *p* > 0.001.

**Figure 4 ijms-24-01568-f004:**
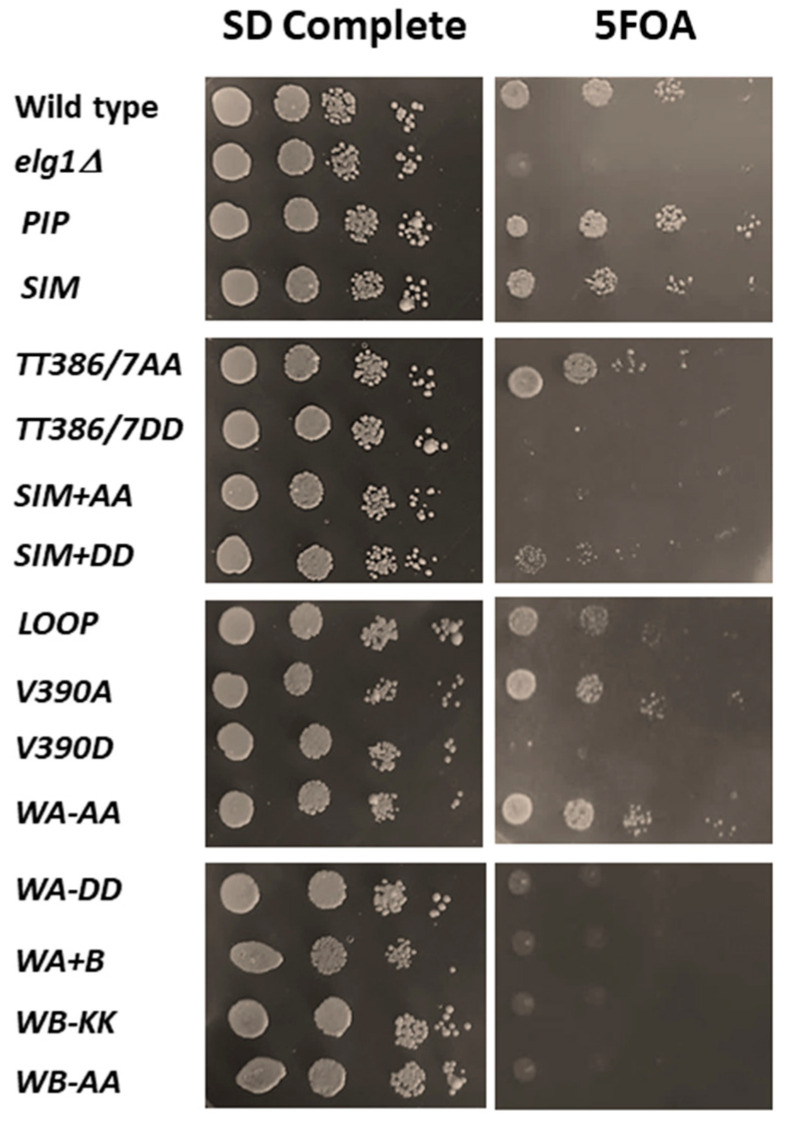
Rescue of the synthetic lethality of *elg1∆ srs2∆.* Diploid stains homozygous for *ELG1* and *SRS2* genomic deletions are kept alive by the presence of a plasmid carrying the *ELG1* and *URA3* genes. We introduced into these strains the *elg1* allele collection on *LEU2*-marked pRS415 vectors. 5-FOA-containing medium selects for cells able to lose the *URA3-ELG1* covering plasmid. The strain described as “Wild type” carried the *ELG1* gene, and the one described as “*elg1Δ*” carried an empty pRS415 vector.

**Figure 5 ijms-24-01568-f005:**
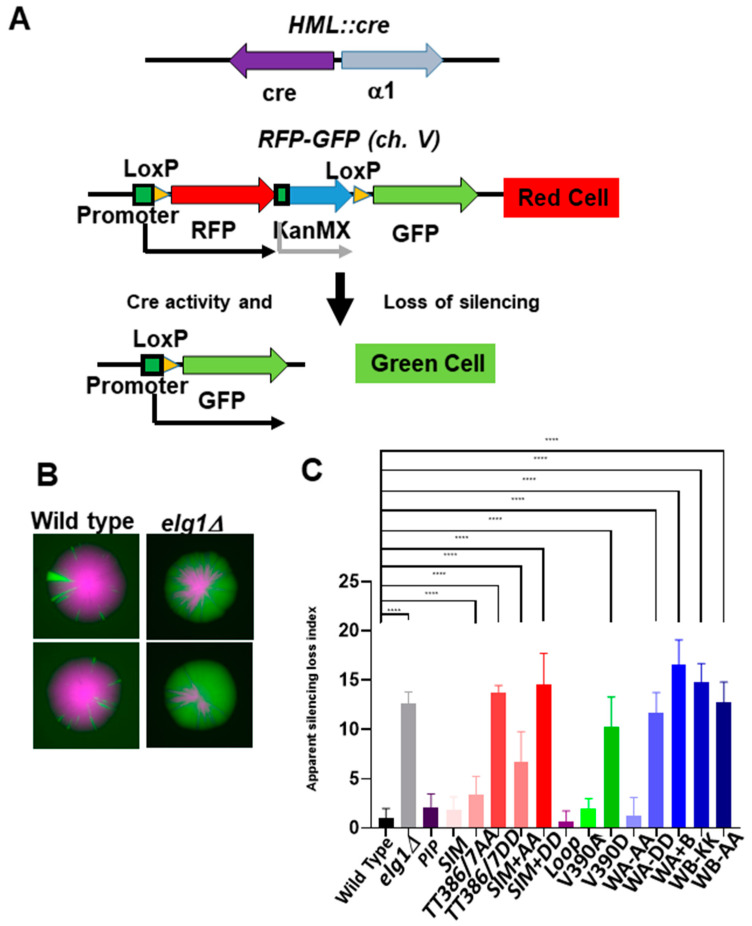
**The apparent silencing-loss rates of the different *elg1* mutations.** (**A**) Schematic description of the CRASH assay. (**B**) Colony sectoring of wildtype and *elg1Δ* alleles. (**C**) Genomic versions of the *elg1* allele collection were created in CRASH-containing strains. Cre activity was quantified using flow cytometry, as described in Materials and Methods. **** *p* < 0.0001 was calculated using ANOVA and Dunnett’s test post-hoc analysis.

**Figure 6 ijms-24-01568-f006:**
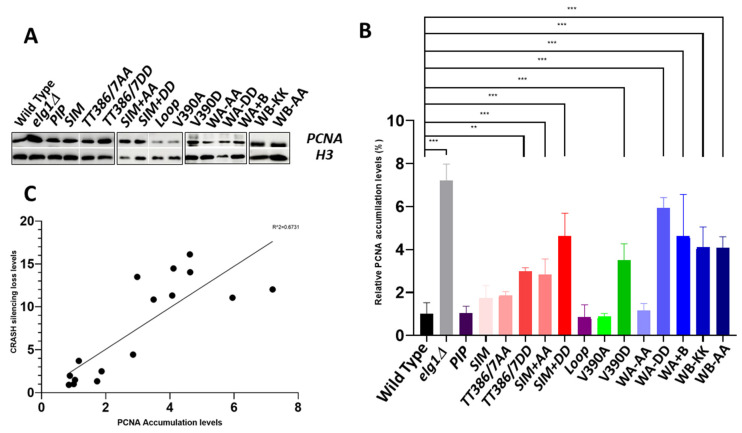
**PCNA chromatin levels for the different *elg1* mutations.** (**A**) A fractionation assay followed by Western blot to detect PCNA levels on the chromatin in the various *elg1* mutants. Histone H3 served as a chromatin-bound- as well as loading control. The full blots are shown as Appendix A. (**B**) Relative levels of silencing loss. At least three experiments were used for quantitation and the error bars represent the standard error of the mean. The graph represents the quantification of Western blots of the relative chromatin-bound PCNA abundance of the different Elg1 mutants in the cell, compared to the relative level of wildtype cells. **, 0.006 > *p* > 0.001; ***, *p* < 0.0001; values were calculated using ANOVA and Dunnett’s test post-hoc analysis. (**C**). **Correlation between PCNA accumulation and the apparent silencing-loss rate.** A correlation is observed between the levels of PCNA and the loss of silencing for each *elg1* mutant as measured by the CRASH assay. The R^2^ is 0.6731 (*p* < 0.0001).

## Data Availability

Not applicable.

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
