# Peer review of "Effects of Defective Unloading and Recycling of PCNA Revealed by the Analysis of ELG1 Mutants"

_ijms, 2023, doi:10.3390/ijms24021568_

Round 1

Reviewer 1 Report

Itzkovich et al. constructed various types of Elg1 mutants in its functional domains or motives and expressed in the cells, and examined their phenotypes, such as sensitivity to DNA damaging agents, mutation rates and defect in silencing. Elg1 is an unloader of PCNA from chromatin, and its deletion leads to accumulation of PCNA on chromatin, causing genome instability. The authors found that each elg1 mutant showed different levels of PCNA accumulation on chromatin and noticed that the extents of their phenotypes on genome stability corelated with the levels of PCNA left on chromatin. Based on these finding, they propose that sole role of Elg1 is timely unloading and recycling of PCNA. Although the authors carefully examined the phenotypes of various Elg1 mutants, more convincing data need to be presented before acceptance in IJMS.

Major points

1) Figure 6A: The western blotting presented is a kind of patchwork made after combining several blottings. Since the actual amounts of PCNA left on chromatin in each mutant are important point of authors’ argument, authors should show the PCNA levels on a single blotting.

2) The authors describe that PCNA detected on chromatin in Elg1 mutants reflects those left unloaded, but it does not necessarily mean so, because difference in the proportion of S-phase in the cell cycle between the mutants affects interpretation of chromatin-bound portion of PCNA. To neglect such a possibility, authors need to show that cell cycle profiles are similar among the mutants by flowcytometry.

3) Figure 6C: I cannot find this figure in the manuscript.

4) From line 314 to 317: The figure numbers are not correctly shown.

5) Title: I don’t feel that the “untimely unloading of PCNA” fits with the results presented by this manuscript. All the mutants appear to have a defect in unloading and thus unloading process was delayed. “Untimely unloading” mutant also includes such one that shows a premature unloading of PCNA from chromatin.

Reviewer 2 Report

In this study, Itzkovich and colleagues aimed to investigate the mechanistic functions of ELG1 in genome stability, particularly, in the PCNA unloading mechanism during DNA replication. For experimental addressing, authors employed a set of ELG1 mutants including, but not limited to PCNA binding mutant, SUMO interaction mutant, hydrophobic loop mutant, and Walker A-B mutants. The selection of such a rich variety of mutants enables authors to address the main question from a wider perspective. Upon chemical induction of DNA replication stress by MMS, ELG1 knockout and some mutant strains displayed increased sensitivity. Further, those mutants displaying increased sensitivity also showed an increased mutation rate. Subsequently, the authors measured the gene-silencing role of ELG1 mutants and revealed a significant loss of gene silencing in certain mutants. Ultimately, a western immunoblotting assay aiming to measure the PCNA protein levels in the chromatin fraction of cells revealed an increased PCNA accumulation in some mutants which also showed gene silencing defects as well as high mutation rates.

The study is investigated in a comprehensive way, however, it suffers from essential shortcomings and a lack of controls. The authors are kindly asked to consider the following major points to improve the manuscript.

1- For the DNA damage sensitivity assay, as MMS stalls the replication fork, it is suggested to perform the assay in G1-S-phase released yeast cells to be able to correctly assess the replication fork stress response associated sensitivity in different ELG1 mutation conditions. Also, as a control, another type of cellular stressor (other than DNA damaging agent) should be tested to clearly identify the involvement of ELG1 mutants in response to replication stress.

2- A confirmation western immunoblotting is suggested for the ELG1 knockout strain, as well as mutants to confirm the loss of ELG1 protein, and expression levels of mutants respectively.

3- How does the cell cycle get affected by the expression of mutants? As it is known that heterochromatin/euchromatin dynamics have a close association with the cell cycle phase. Gene silencing findings in Fig.5 should be further evaluated by considering the cell cycle distribution of mutants.

4- Findings in Fig.6 should further get normalized with the expression levels of corresponding ELG1 wildtype and mutant proteins.

5- Have the authors tested what could be the functional consequence of PCNA accumulation? In a conventional way of thinking, there could be three reasons caused by and/or resulting in increased PCNA accumulation on chromatin: Replication stress-induced stall that prevents dissociation, defect in loading/unloading machinery, and increased DNA replication efficiency. In this study, ELG1 is a known PCNA unloading machinery component, however, the conclusion is not clear and not supported by a functional assay (e.g. DNA fiber assay) whether the ELG1 knockout and mutants cause alteration in DNA replication proficiency and efficiency. Authors are strongly encouraged to consider measuring a functional output to strengthen the story.

Author Response

Pls see the attachment

Round 2

Reviewer 1 Report

In the revised manuscript, the authors addressed all my concerns reasonably, and now I recommend it for publication.

Author Response

We thank the reviewer again.

Reviewer 2 Report

I thank the authors for improving the manuscript by considering the suggestions. However, I would again strongly encourage the authors to reconsider major-point-6 to further increase the significance of the story. The currently employed experimental approaches to investigate the functional consequences of PCNA accumulation are addressing phenotypical and systemic responses (e.g. cell death, DNA damage sensitivity) as the authors also mentioned, but that is not explaining how the DNA replication mechanism is affected by this accumulation. 

Author Response

We thank the reviewer for her/his comment, which we  think  is about point 5, to do DNA fiber experiments. Although we agree that this is a good idea, it is far beyond the scope of this paper. Our lab has never done such experiments, and it will take quite a while to organize them and execute them. We think we have provided many insights about the phenotypes of Elg1 and the biological effects of failing to unload PCNA, and that warrant their publication. Thus, DNA fiber (or similar) experiments will have to wait for the moment.

Round 3

Reviewer 2 Report

Actually, assessment of the replication proficiency is at the heart of the scope if PCNA is the core of the study. However, it is understandable that establishment and optimization of such a technique would require considerable amount of time and resources. Therefore, in the current version of the manuscript, it is necessary that the statements about DNA replication in the sections of abstract and results, should be toned down.

Author Response

We have deleted the reference to DNA replication in the abstract. We went through the Results and the Discussion sections, but we could not find any claim related to DNA replication to tone down.